# Synthesis and Properties of Optically Transparent Fluoro-Containing Polyimide Films with Reduced Linear Coefficients of Thermal Expansion from Organo-Soluble Resins Derived from Aromatic Diamine with Benzanilide Units

**DOI:** 10.3390/ma15186346

**Published:** 2022-09-13

**Authors:** Xi Ren, Hanli Wang, Xuanzhe Du, Haoran Qi, Zhen Pan, Xiaolei Wang, Shengwei Dai, Changxu Yang, Jingang Liu

**Affiliations:** 1School of Materials Science and Technology, China University of Geosciences, Beijing 100083, China; 2Shandong Huaxia Shenzhou New Material Co., Ltd., Zibo 256401, China

**Keywords:** polyimide, trifluoromethyl, solution processability, high modulus, linear coefficients of thermal expansion

## Abstract

Wholly aromatic polyimide (PI) films with good solution processability, light colors, good optical transparency, high storage modulus, and improved heat resistance were prepared and characterized. For this purpose, a multi-component copolymerization methodology was performed from a fluoro-containing dianhydride, 4,4′-(hexafluoroisopropylidene)diphthalic anhydride (6FDA), a rigid dianhydride, 3,3′,4,4′-biphenyltetracarboxylic dianhydride (BPDA), and a fluoro-containing diamine, 2,2′-bis(trifluoromethyl)-4,4′-bis [4-(4-amino-3-methyl)benzamide]biphenyl (MABTFMB). One homopolymer, FPI-1 (6FDA-MABTFMB), and five copolymers, FPI-2~FPI-6, containing the BPDA units from 10 mol% to 50 mol% in the dianhydride moieties, were prepared, respectively. The derived PI resins showed good solubility in the polar aprotic solvents, such as N-methyl-2-pyrrolidone (NMP) and N,N-dimethylacetamide (DMAc). The flexible PI films obtained by the solution casting procedure showed good optical properties with the transmittances higher than 74.0% at the wavelength of 450 nm. The PI films exhibited excellent thermal properties, including 5% weight loss temperatures (T_5%_) over 510 °C, together with glass transition temperatures (T_g_) over 350.0 °C according to the peak temperatures of the loss modulus in dynamical mechanical analysis (DMA) measurements. The FPI-6 film also showed the lowest linear coefficient of thermal expansion (CTE) value of 23.4 × 10^−6^/K from 50 to 250 °C according to the thermomechanical analysis (TMA) measurements, which was obviously lower than that of FPI-1 (CTE = 30.6 × 10^−6^/K).

## 1. Introduction

Colorless or pale-colored and optically transparent polyimide (PI) films are well-known in advanced optoelectronic applications for their excellent comprehensive properties [1,2,3]. After several decades’ basic and applied research, colorless PI films have been gradually achieving commercial applications in flexible electronics, flexible solar cells, flexible printed circuit boards, and other high-tech fields [4]. However, the methodologies of structural modification for simultaneously achieving high optical transparency and thermal stability, especially dimensional stability at elevated temperatures in colorless PI films, are often mutually contradictory. For example, in order to endow the common golden PI films’ colorless features, various substituents with low conjugated structures and low intra- or intermolecular interactions, such as flexible groups, alicyclic units, or latent bulky alkyl or aryl groups, have to be introduced into the PIs in order to prohibit the charge transfer (CT) interactions from the diamine to the dianhydride moieties [5,6,7]. These modifications inevitably sacrifice the intrinsic thermal stability of the pristine PI films. Thus, compromise between the thermal and optical properties for colorless PI films has to be adopted in practical applications [8]. This usually leads to very narrow modification windows for the research and development (R&D) of colorless PI films. For example, in commercial applications, the colorless PI films based on the copolymerization of 4,4′-(hexafluoroisopropylidene)diphthalic anhydride (6FDA), rigid-rod 3,3′,4,4′-biphenyltetracarboxylic dianhydride (BPDA), and rigid-rod 2,2′-bis(trifluoromethyl)benzidine (TFMB) have been paid increasing attention because the fluoro-containing units in 6FDA and TFMB could endow the colorless PI films’ desirable optical properties, while the rigid biphenyl units in BPDA and TFMB could efficiently decrease the linear coefficients of thermal expansion (CTE) and increase the modulus of the derived colorless PI films [9,10]. A molar ratio over 50% for the BPDA dianhydride in the total dianhydride moieties usually has to be adopted so as to fulfill the requirements of advanced optoelectronic applications. Such high molar ratios for the rigid BPDA components in colorless PI films greatly deteriorates the solubility of the colorless PIs in organic solvents; thus, making it quite difficult to fabricate the films in the forms of preimidized CPI solutions. In practice, a two-stage manufacturing procedure via the soluble poly(amic acid) (PAA) precursors has to be used. The severe imidization conditions from PAA precursors to the final colorless PIs, such as dehydrating temperatures as high as 350 °C in the inert gas environments, bring so many technical barriers for the continuous manufacturing of colorless PI films. Thus, R&D of novel colorless or light-colored PI systems with both good solution processability in preimidized forms at relatively lower temperatures and the lowest possible CTEs and high modulus have attracted increasing attentions in recent years [11,12,13,14,15].

Among the various modifications in the literature, the incorporation of rigid-rod amide or benzanilide units into the colorless PI films has been becoming one of the most attractive methodologies for achieving the best trade-off among various properties for the polymers [16,17]. This is mainly due to the intrinsic characteristics for the amide groups, including the high polarity, high chemical bonding energies, and great potentiality forming the strong intermolecular and intramolecular hydrogen bonds. However, the amide or benzanilide units usually showed certain absorptions to the visible light. Thus, in the practical modification, these groups are usually cooperated with other functional groups with lower optical absorptions. For example, very recently, Ahn and coworkers reported the preparation of optically transparent, optically colorless PI films with excellent mechanical strength and folding reliability using 6FDA, BPDA, and trifluoromethyl side chains containing amine-terminated amide oligomer (AAO) as the starting monomers by a chemical dehydration procedure [18]. The colorless PI films were fabricated at the relatively lower processing temperature of 250 °C and the resulting colorless PI films showed the highest tensile modulus of 8.4 GPa, glass transition temperatures (T_g_) higher than 360 °C, optical transmittance around 90%, and yellow indices below three. Thus, good compromise between the heat resistance, mechanical properties, and optical properties was successfully achieved for the derived CPI films. Hasegawa and coworkers reported a para-amide-linked diamine, 2,2′-bis(trifluoromethyl)-4,4′-bis [4-(4-aminobenzamide)]biphenyl (AB-TFMB) and the derived colorless PI films [19]. The diamine was designed for achieving good trade-off among various properties of the afforded colorless PI films and experimental results well proved the successful molecular design. Both the semi-alicyclic colorless PI films derived from AB-TFMB and alicyclic dianhydride, such as cyclobutanetetracarboxylic dianhydride (CBDA), and derived from aromatic dianhydride, such as 6FDA, showed reduced CTE values (CTE = 11.7 × 10^−6^/K for PI_CBDA/AB-TFMB_ and 33.6 × 10^−6^/K for PI_6FDA/AB-TFMB_ in the temperature range of 50 to 250 °C, respectively). In our previous study, the semi-aliphatic colorless PI films were also prepared from methyl-substituted 4,4′-diaminobenzanilide and alicyclic dianhydrides [20,21]. Although the derived colorless PI films exhibited good solution processability and good optical transmittance, the CTE values still needed to be further reduced.

In the current work, a series of wholly aromatic light-colored fluoro-containing PI (FPI) films were designed to possess good solution processability and comprehensive properties for advanced optoelectronic applications. A methyl-substituted aromatic diamine analogous to AB-TFMB, 2,2′-bis(trifluoromethyl)-4,4′-bis [4-(4-amino-3-methyl)-benzamide]biphenyl (MABTFMB) was copolymerized with 6FDA and BPDA. Influences of the methyl and trifluoromethyl substituents in the side chains and the rigid amide and biphenyl linkages in the main chains on the processability, thermal, and optical properties of the derived FPI films were investigated.

## 2. Materials and Methods

### 2.1. Materials

4,4′-(Hexafluoroisopropylidene)diphthalic anhydride (6FDA) and 3,3′,4,4′-biphenyltetracarboxylic dianhydride (BPDA) were purchased from China Tech (Tianjin, China) Chemical Co., Ltd. (Tianjin, China). They were pretreated at 180 °C in vacuo for 10 h. 2,2′-Bis(trifluoromethyl)-4,4′-bis [4-(4-amino-3-methyl)benzamide]biphenyl (MABTFMB) was mass-produced in house and recrystallized from ethanol prior to use. Anhydrous N,N-dimethylacetamide (DMAc), N-methyl-2-pyrrolidinone (NMP), N,N-dimethylformamide (DMF), and γ-butyrolactone (GBL) with water contents lower than 50 ppm were all purchased from Sinopharm Co., Ltd. (Shanghai, China) and used directly.

### 2.2. PI Synthesis

One homopolymer, FPI-1 (6FDA-MABTFMB), and five copolymers, FPI-2−FPI-6, containing the BPDA from 10 mol% to 50 mol% in the dianhydride moieties, were prepared, respectively, according to the formula shown in Table 1. The synthesis pathway could be presented by the preparation of FPI-6. MABTFMB (11.7306 g, 20 mmol) and the anhydrous NMP (57.4 g) were added in a 250 mL glass flask with a cold water bath. The reaction system was stirred at 5~10 °C for 20 min under nitrogen to give a pale-yellow solution. Then, 6FDA (4.4424 g, 10 mmol) and BPDA (2.9422 g, 10 mmol) were added into the flask. The total solid content of the reaction mixture was controlled to be 25 wt%. After stirring at 5~10 °C for 3 h, the cold water bath was removed and the reaction mixture naturally warmed to room temperature (25 °C). The reaction system was continuously stirred at room temperature for another 22 h to afford a highly viscous solution. Then, the dehydrating agent of acetic anhydride (Ac_2_O) (10.2 g, 100 mmol) and the catalyst of pyridine (6.3 g, 80 mmol) were added to the reaction solution, after which the reaction mixture was stirred for an additional 24 h. The obtained FPI-6 solution was poured into the precipitating agents of 500 mL ethanol solution (75 vol%). The precipitated FPI-6 resin was collected and dried at 100 °C in vacuum overnight. Yield: 17.7 g (96%).

Then, 15.0 g of the dried FPI-6 was added into 85.0 g of the NMP at a solid content of 15 wt%. The mixture was stirred at 25 °C for 24 h. The afforded FPI-6 solution was purified by passing through a 0.45 μm filter. Then, the FPI-6 solution was cast onto the clean glass. The glass substrates were then heated in an oven according to the program of 80 °C for 2 h, 120 °C for 1 h, 180 °C for 1 h, and 250 °C for 1 h. The free-standing FPI-6 film was then peeled off the substrates by immersing them into water for several hours. FTIR (cm^−1^): 1778, 1717, 1674, 1493, 1369, 1319, 1103, and 721.

The other resins and films were prepared similarly. The 6FDA/BPDA molar ratios were controlled to be 100:0 for FPI-1, 90:10 for FPI-2, 80:20 for FPI-3, 70:30 for FPI-4, and 60:40 for FPI-5, respectively.

FPI-1. FTIR (cm^−1^): 1786, 1721, 1678, 1493, 1369, 1319, 1103, and 721. FPI-2. FTIR (cm^−1^): 1782, 1721, 1674, 1493, 1369, 1319, 1103, and 721. FPI-3. FTIR (cm^−1^): 1782, 1721, 1674, 1493, 1369, 1319, 1103, and 721. FPI-4. FTIR (cm^−1^): 1782, 1721, 1674, 1493, 1369, 1319, 1103, and 721. FPI-5. FTIR (cm^−1^): 1782, 1721, 1674, 1493, 1369, 1319, 1103, and 721.

### 2.3. Characterization

The weight average molecular weight (M_w_) and number average molecular weight (M_n_) of the PI resins dissolved in NMP containing 0.025 mol/L phosphoric acid were detected using a gel permeation chromatography (GPC) system (Shimadzu, Kyoto, Japan). The Iraffinity-1S FT-IR spectrometer (Shimadzu, Kyoto, Japan) and the U-3210 spectrophotometer (Hitachi, Tokyo, Japan) were used to measure the Fourier transform infrared (FTIR) and ultraviolet-visible (UV-Vis) spectra of the PI films, respectively. Wide-angle X-ray diffraction (XRD) was performed on a D/max-2500 X-ray diffractometer (Rigaku, Tokyo, Japan). The CIE (International Commission on Illumination) Lab optical parameters of the PI films were measured using a color i7 spectrophotometer (X-rite, Grand Rapids, MI, USA) at a thickness of 50 μm. The thermal properties of the PI films were separately evaluated by thermogravimetric analysis (TGA) via a Q50 thermal analysis system (New Castle, DE, USA), by dynamic mechanical analysis (DMA) via a TA-Q800 thermal analysis system (New Castle, DE, USA), and thermo-mechanical analysis (TMA) via a TMA402F3 thermal analysis system (NETZSCH, Selb, Germany). The heating rates were 20 °C/min, 5 °C/min, and 5 °C/min for TGA, DMA, and TMA analyses, respectively and all the analyses were performed in nitrogen. The coefficients of linear thermal expansion (CTE) values of the PI films were recorded in the range of 50–250 °C.

Solubility of the PI resin in various solvents was measured by preparing a PI/solvent mixture at a solid content of 10 wt%. The mixture was stirred for 24 h at room temperature. The solubility was determined visually as three grades: completely soluble (++), partially soluble (+−), and insoluble (−).

## 3. Results

### 3.1. PI Synthesis and Characterization

The fluoro-containing PI films were prepared via a two-stage chemical dehydration procedure, as shown in Figure 1. As mentioned in the Introduction, the fluoro-containing colorless PI system based on 6FDA, BPDA, and TFMB represented the standard colorless PI films for commercial applications due to the good trade-off among various properties. However, the limited solubility of the colorless PI system, due to the high molar contents of BPDA in the dianhydride moiety, made it only capable of being processed via the thermal imidization procedure via the soluble PAA intermediates. The colorless PI system based on 6FDA, BPDA, and AB-TFMB also has to face the same problem. In the current work, the methyl substituents were introduced to the AB-TFMB diamine to afford MABTFMB, so as to increase the solubility of the derived FPI resins. As expected, all the polycondensation systems maintained homogeneous and flowable states during the chemical imidization processing. The FPI resins were successfully obtained by precipitating the soluble PI solutions into the poor solvents.

The molecular weights of the FPI resins were evaluated first and the results are shown in Table 2. All the FPI resins showed a number of molecular weights (M_n_) higher than 3.0 × 10^4^ g/mol, indicating the good polymerization of the MABTFMB diamine. Although the electron densities in the amino groups might be reduced by the para-substituted amide units with an electron-withdrawing feature, the electron-donating ortho-substituted methyl groups increased the electron densities in the amino groups. Thus, the amino groups exhibited good reactivity in the nucleophilic acyl substitution reactions when the nitrogen in the amino groups attacked the carbon of the carbonyl groups in the dianhydride moiety. In addition, incorporation of BPDA components into the FPIs did not obviously affect the M_n_ values of the polymers, which might be due to the similar reactivity of the BPDA and 6FDA dianhydrides. Although the M_n_ values of the FPI resins were not apparently affected by the incorporation of BPDA components, the solubility of the FPI resins in organic solvents was found to be quite sensitive with the contents of BPDA units in the polymers. It could be seen from the solubility tests results shown in Table 2 that with the increasing of the BPDA components in the FPIs, the solubility gradually deteriorated. All the FPI resins showed good solubility in polar aprotic solvents, such as NMP and DMAc, at a solid content of 10 wt%. It demonstrates that although the rigid benzanilide and biphenyl units were incorporated into the polymer chains, the polymers showed loose molecular chain packing and low crystalline states, as evidenced by the XRD measurements shown in Figure 2. The amorphous nature for the current samples might be the synergistic effects of the bulky hexafluoroisopropylidene in the dianhydride moiety and the trifluoromethyl and methyl substituents in the diamine moiety. FPI-1 was also soluble in DMF and GBL and partially soluble in chloroform and THF at room temperature. However, FPI-5 and FPI-6 were all not soluble in these solvents. The deterioration of the solubility for the FPI resins with high BPDA contents could naturally be ascribed to the rigid molecular skeleton in BPDA.

The good solubility of the current FPI resins in NMP and DMAc made it possible to fabricate the CPI films via the preimidized FPI solutions, as shown in Figure 3. The FPI resins were first dissolved into NMP at a solid content of 15wt%. Viscous FPI solutions were then obtained after stirring at room temperature overnight. It could be seen from the figure that FPI-1–FPI-4 showed a homogeneous status after standing for several hours at room temperature. However, obvious bubbles were observed for FPI-5 and FPI-6. It demonstrates that although all the FPI resins were soluble in NMP, the solubility might be different. Nevertheless, homogeneous solutions were also obtained for FPI-5 and FPI-6 after defoaming in vacuo for half an hour. At last, flexible and tough FPI films with pale-yellow colors were obtained by thermally baking the cast FPI solutions at relatively low temperatures.

Figure 4 exhibits the FTIR spectra of the FPI films. The characteristic absorptions of the imide rings were all observed. For the imide carbonyl groups, stretching vibrations at the wavenumbers of 1778 cm^−1^ (asymmetric) and 1717 cm^−1^ (symmetric) and in-plane bending vibrations at the wavenumbers of 721 cm^−1^ were all clearly detected. For the imide C-N bonds, stretching vibrations at 1369 cm^−1^ were observed. Meanwhile, absorptions of C=O in amide bonds at 1674 cm^−1^ and C=C in phenyl rings at 1493 cm^−1^ were also detected for all of the polymers. As for the C-F bonds, stretching vibrations of C-F bonds in the aliphatic hexafluoroisopropylidene (-C(CF_3_)_2_) units in 6FDA were all detected at 1103 cm^−1^, whereas asymmetric stretching vibrations of C-F bonds attached to benzene rings in the MABTFMB units were observed at 1319 cm^−1^. The FTIR spectra indicated the successful transition from the monomers to the target FPI films.

### 3.2. Optical Properties

It can be observed from Figure 2 that the FPI films showed low colors and good transparency. The optical properties, including the optical transmittance in the ultraviolet and visible lights region and the color parameters of the FPI films, were quantitatively investigated and the results are tabulated in Table 3. Figure 5 illustrates the UV-Vis spectra of the FPI films. The FPI films exhibited cutoff wavelengths (λ_cut_) in the range of 340–367 nm. The optical transmittances of the FPI films were in the range of 60.3–82.1% at the wavelength of 400 nm (T_400_) and 74.0–86.4% at the wavelength of 450 nm (T_450_), respectively. The FPI films exhibited decreasing T_400_ and T_450_ values with the order of FPI-1 > FPI-2 > FPI-3 > FPI-4 > FPI-5 > FPI-6. Apparently, with the increasing contents of BPDA components in the FPI films, the optical transparency gradually deteriorated. This is mainly attributed to the gradually enhanced CT interactions in the FPI films due to the increasing highly-conjugated BPDA contents in the polymers [22,23,24].

The CIE Lab color parameters of the FPI films are listed in Table 3, and the three-dimensional CIE Lab plots are shown in Figure 6. All the FPI films showed high lightness (L*) over 94.0, and negative a* and positive b* values, indicating the highly transparent and green and yellow nature in the colors. The yellow index (b*) values of the FPI films increased from 1.14 (FPI-1) to 2.62 (FPI-6) with the increasing of the BPDA contents in the polymers. This is also due to the highly conjugated molecular skeletons of the BPDA dianhydride. Nevertheless, the b* values were all lower than 3.0, indicting the low colors of the FPI films. This is mainly due to the existence of the high contents of highly electronegative trifluoromethyl units both in the dianhydride and diamine moieties. The intra- and intermolecular CT interactions in the molecular chains of the FPIs were apparently weakened. In addition, the FPI films exhibited low haze values in the range of 0.38–2.29%. FPI-6, with the highest BPDA content, showed the highest haze value of 2.29%. This might be due to the somewhat higher degree of turbidity in the film caused by the reduced solubility of the starting FPI-6 resin. In summary, the current FPI films showed highly ranked optical transmittance and CIE Lab parameters among the wholly aromatic colorless PI films reported in the literature [25,26]. This is helpful for the applications in optoelectronic fields.

### 3.3. Thermal Properties

Synergistic effects of the rigid benzanilide and biphenyl units and the flexible methyl and trifluoromethyl groups on the thermal properties of the FPI films were systematically investigated and the corresponding thermal data are listed in Table 4. First, Figure 7 shows the TGA and derivative TGA (DTG) plots of the FPI films. All the FPI films exhibited good thermal stability before 500 °C in nitrogen. With the increasing test temperatures, the FPI films revealed 5% weight loss temperatures (T_5%_) in the range of 513.7–524.4 °C. Basically, incorporation of the rigid BPDA components in the FPI films slightly increased the T_5%_ values of the polymers. For example, FPI-6, with the highest BPDA contents, showed the T_5%_ value of 524.4 °C, which was more than 10 °C higher than that of FPI-1. According to the DTG plots, the current FPI films showed clear two-stage thermal decomposition behaviors. The first stage occurred in the temperature range of 525–535 °C, which could be attributed to the decomposition of side chains in the polymers, such as methyl and trifluoromethyl groups. The second decomposition detected in the temperature range of 580–595 °C could be ascribed to the decomposition of the main chains of the polymers. At last, all the FPI films left more than 50wt% of their original weights at 750 °C, indicating the excellent thermal stability of the polymers.

Figure 8 shows the relationships between the modulus, including the storage modulus (E′) and loss modulus (E″) of the FPI films, with the test temperatures revealed by DMA measurements. According to Figure 8a and Table 4, the initial E′ values of the FPI films at 50 °C increased with the order of FPI-1(1.21 GPa) < FPI-2(3.56 GPa) < FPI-3(3.91 GPa) < FPI-4(4.25 GPa) < FPI-5(4.53 GPa) < FPI-6(9.87 GPa). The copolymer films even maintained E′ values higher than 1.0 GPa at 300 °C. Thus, the storage modulus of the FPI films gradually increased with the increasing BPDA contents in the polymers, especially when the molar ratio of BPDA reached 50% in the dianhydride moiety. It has been well-established in the literature that the PI films derived from the rigid BPDA dianhydride usually exhibited high storage moduli both at room temperature and at high temperatures [27,28,29]. In the current work, besides the contribution of rigid BPDA components to the high E′ values of the films, the rigid benzanilide units in the MABTFMB moiety also endow the FPI films’ high modulus [30,31,32]. The high E′ values of the current FPI films, especially for FPI-6, reflected the strong and hard nature of the film, which might be beneficial for applications in advanced optoelectronic fields, such as the cover window films for flexible display devices [33].

As for the loss modulus of the FPI films, all the polymers maintained nearly constant E″ values before 320 °C. When the test temperatures were close to the T_g_ of the polymers, the E″ plots raised first and then dropped rapidly. The peak temperatures of the E″ plots were recorded as the T_g_ values of the FPI films. It can be seen from Figure 8a and Table 4 that all FPI films showed similar T_g_ values in the range of 357–367 °C. The high T_g_ values of the FPI films, on one hand, could be ascribed to the effects of the rigid benzanilide and biphenyl units in the polymers. On the other hand, the ortho-substituted methyl groups prohibited the free motion of the molecular chain segments in the FPIs, thus efficiently increasing the T_g_ values [34]. The high-T_g_ feature of the FPI films can also be deduced from the tan delta plots of the FPI films shown in Figure 8b. As the ratio of the E″ and E′, the peak temperatures of tan delta plots could also represent the T_g_ values of the test samples. None of the FPI films showed peaks in the plots before 400 °C, indicating the excellent high-temperature viscoelastic stability of the current polymers.

At last, the dimensional stability of the FPI films at elevated temperatures was evaluated by TMA measurements. Figure 9 shows the dimension change–temperature curves of the FPI films. All the FPI films exhibited good dimensional stability before 320 °C and drastic shrinkage in the dimensions of the test samples occurred in the temperature range of 320–400 °C, indicating the ordered arrangements of the molecular chains of the polymers at high temperatures. When the arrangements of the molecular chains finished, the dimensions of the samples sharply expanded after 400 °C. In order to compare the high-temperature dimensional stability of the FPI films, the linear coefficients of thermal expansion (CTE) values of the FPI films in the temperature range of 50–250 °C were recorded. According to the CTE data shown in Table 4, the current FPI films basically ranked as the low-CTE polymers in the families of optical polymer materials [35]. Considering that the current FPIs possessed good solution processability, the low-CTE values around 30 × 10^−6^/K for the FPI films might be quite valuable for their applications in high-tech areas. The FPI films exhibited decreasing CTE values with the order of FPI-1 > FPI-2 > FPI-3 > FPI-4 > FPI-5 > FPI-6, which was in good consistence with the increasing contents of BPDA components in the polymers. For example, the FPI-6 film showed the lowest CTE value of 23.4 × 10^−6^/K, which was obviously lower than that of FPI-1 (CTE = 30.6 × 10^−6^/K). This is also due to the incorporation of the rigid BPDA components. In addition, it is worth noting that the CTE value of the FPI-6 film was relatively near that of the copper substrate (CTE = 17.0 × 10^−6^/K), indicating that the FPI film might be a good candidate for applications in flexible copper clad laminates (FCCL).

## 4. Conclusions

Functional PI films with good solution processability in preimidized forms, low colors, high optical transmittance, high storage moduli, and low CTE values were designed and developed. Good balance was achieved among the above-mentioned properties via the copolymerization of 6FDA, BPDA, and a newly developed MABTFMB diamine. The derived FPI-6 film with a molar ratio of 50% for the BPDA components in the dianhydride moiety showed the best comprehensive properties, including a T_g_ of 367 °C, a T_450_ value of 74.0%, a b* value of 2.62, a haze value of 2.29%, a E′ value of 9.87 GPa at 50 °C, and a CTE value of 23.4 × 10^−6^/K in the temperature range of 50–250 °C. The tensile properties of the FPI films, including the tensile strength, tensile modulus, and elongations at break were not evaluated in the current work because these data are highly sensitive to the film-forming conditions [36]. The tensile properties of the FPI films manufactured by the industrial biaxially stretching procedures will be reported in our next work. In summary, the good comprehensive properties of the FPI films might endow them good performance in advanced optical and electronic applications.

## Figures and Tables

**Figure 1 materials-15-06346-f001:**
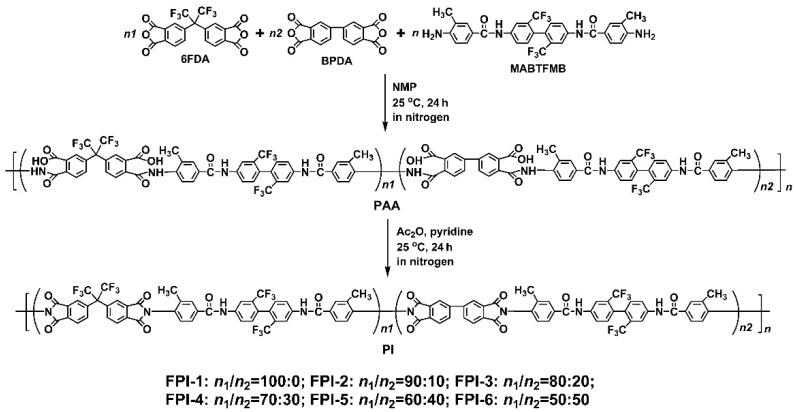
Preparation of organo-soluble FPI resins.

**Figure 2 materials-15-06346-f002:**
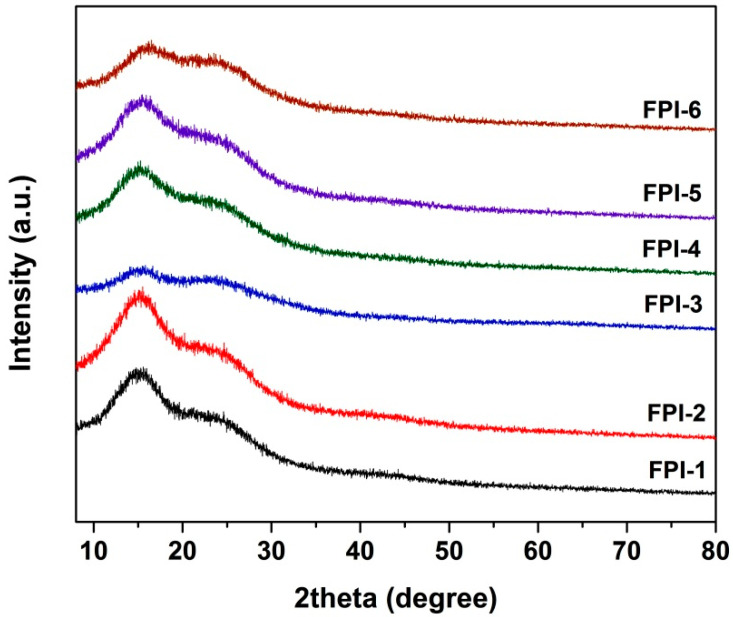
XRD spectra of FPI resins.

**Figure 3 materials-15-06346-f003:**
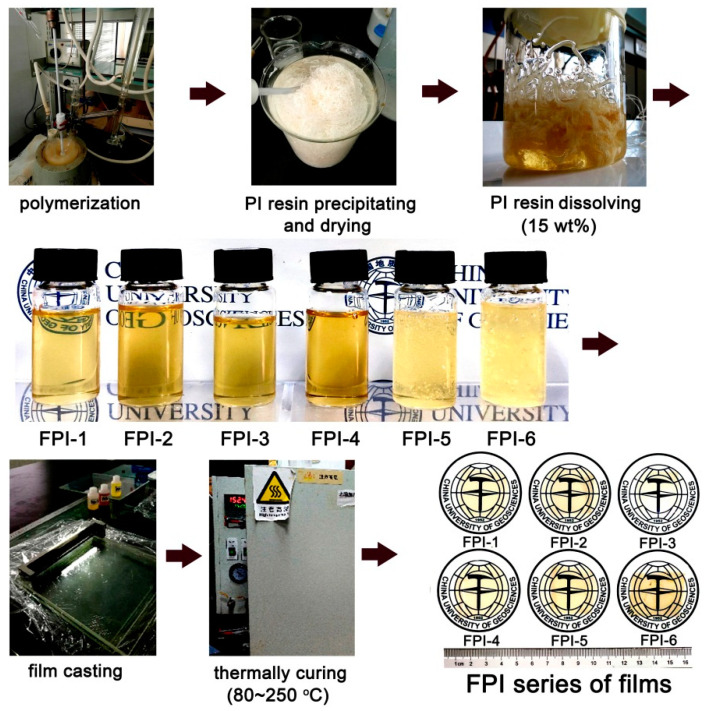
Preparation procedures of the FPI films.

**Figure 4 materials-15-06346-f004:**
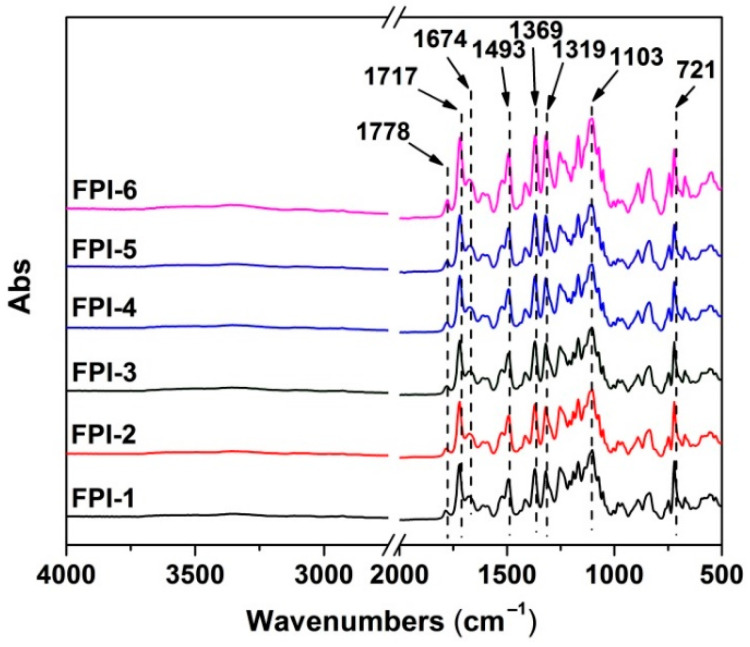
FTIR spectra of FPI films.

**Figure 5 materials-15-06346-f005:**
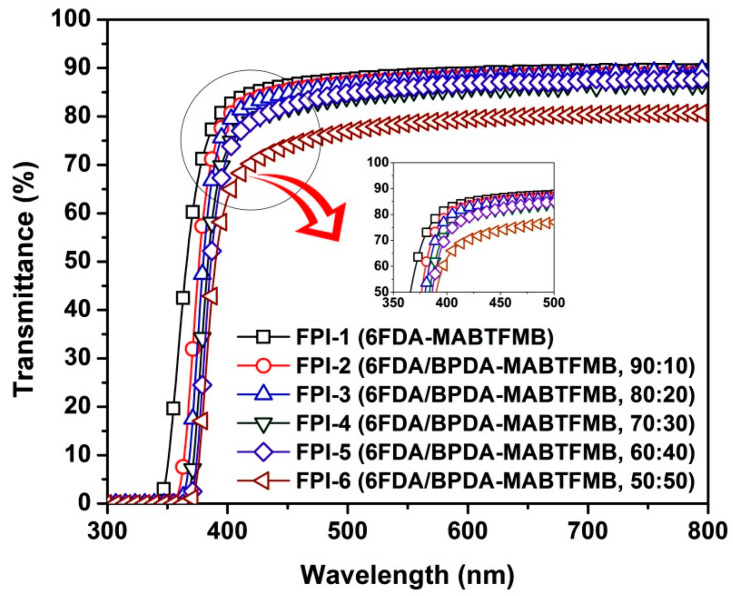
UV-Vis spectra of FPI films.

**Figure 6 materials-15-06346-f006:**
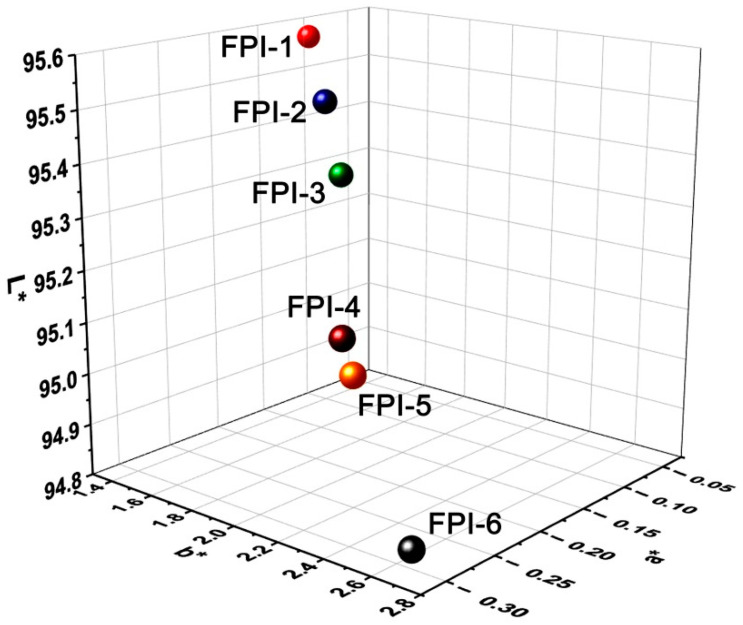
Three-dimensional CIE Lab color parameters of the FPI films.

**Figure 7 materials-15-06346-f007:**
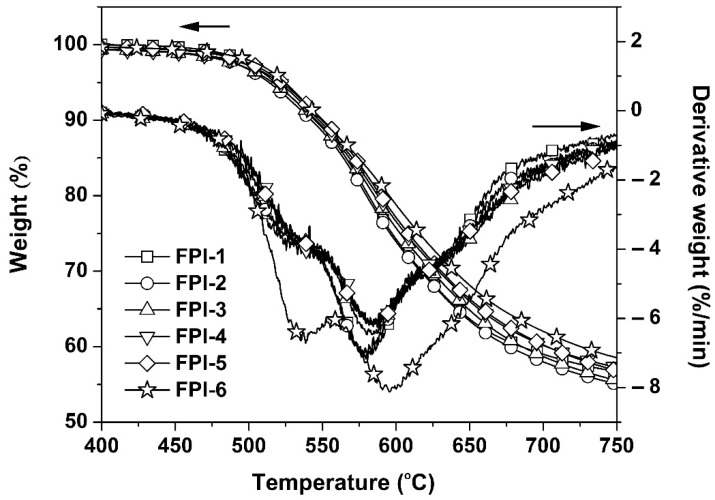
TGA plots of FPI films.

**Figure 8 materials-15-06346-f008:**
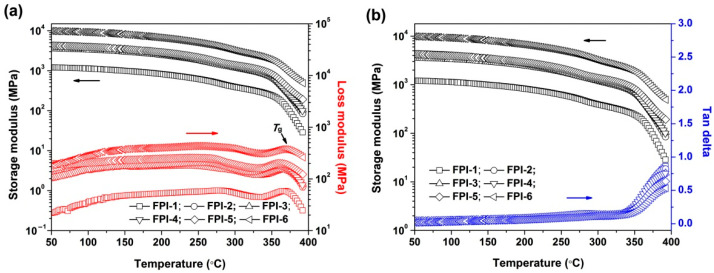
DMA plots of FPI films: (**a**) storage modulus and loss modulus; (**b**) storage modulus and tan delta.

**Figure 9 materials-15-06346-f009:**
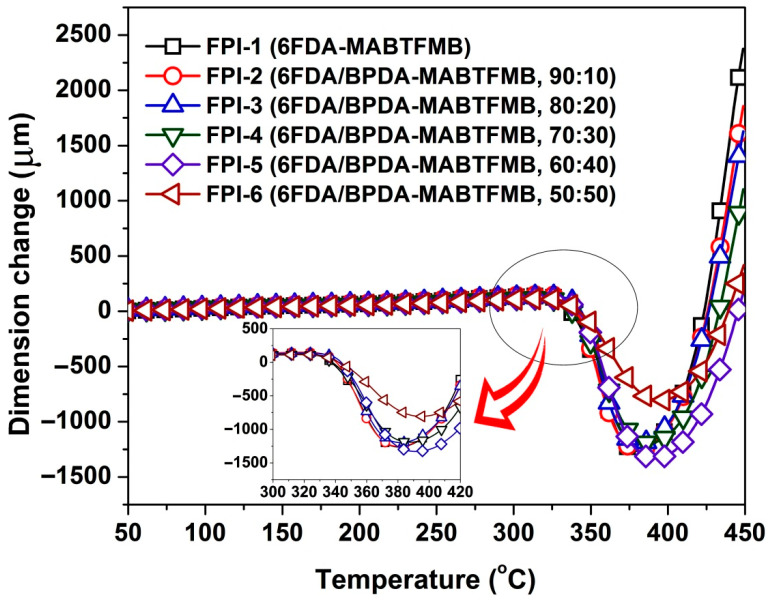
TMA plots of FPI films.

**Table 1 materials-15-06346-t001:** Formula for the FPI synthesis.

PI	6FDA (g, mmol)	BPDA (g, mmol)	MABTFMB (g, mmol)	NMP (g)	Ac_2_O (g, mmol)	Pyridine (g, mmol)
FPI-1	8.8848, 20	0	11.7306, 20	61.8	10.2, 100	6.3, 80
FPI-2	7.9963, 18	0.5884, 2	11.7306, 20	60.9	10.2, 100	6.3, 80
FPI-3	7.1078, 16	1.1769, 4	11.7306, 20	60.0	10.2, 100	6.3, 80
FPI-4	6.2194, 14	1.7653, 6	11.7306, 20	59.1	10.2, 100	6.3, 80
FPI-5	5.3309, 12	2.3538, 8	11.7306, 20	58.2	10.2, 100	6.3, 80
FPI-6	4.4424, 10	2.9422, 10	11.7306, 20	57.4	10.2, 100	6.3, 80

**Table 2 materials-15-06346-t002:** Molecular weights and solubility of FPI resins.

PI	M_n_ ^a^ (g/mol, 10^4^)	M_w_ ^a^ (g/mol, 10^4^)	PDI ^a^	Solubility ^b^					
				NMP	DMAc	DMF	CHCl3	THF	GBL
FPI-1	4.87	9.66	1.98	++	++	++	+−	+−	++
FPI-2	4.05	7.81	1.93	++	++	++	+−	+−	+−
FPI-3	3.92	7.38	1.88	++	++	++	−	−	+−
FPI-4	4.70	9.48	2.02	++	++	++	−	−	+−
FPI-5	4.57	9.01	1.97	++	++	+−	−	−	−
FPI-6	4.82	9.65	2.00	++	++	+−	−	−	−

^a^ M_n_: number average molecular weight; M_w_: weight average molecular weight; ^b^ ++: soluble; +−: partially soluble; −: insoluble; CHCl_3_: chloroform; THF: tetrahydrofuran; GBL: γ-butyrolactone.

**Table 3 materials-15-06346-t003:** Optical properties of the FPI films.

Samples	λ_cut_ ^a^ (nm)	T_400_ ^b^ (%)	T_450_ ^b^ (%)	L* ^c^	a* ^c^	b* ^c^	Haze (%)
FPI-1	340	82.1	86.4	95.54	−0.06	1.14	0.38
FPI-2	352	79.8	85.4	95.47	−0.15	1.63	0.58
FPI-3	358	78.4	84.7	95.34	−0.16	1.75	0.61
FPI-4	363	73.3	81.8	95.14	−0.27	2.24	1.20
FPI-5	366	72.0	82.3	95.09	−0.28	2.33	1.21
FPI-6	367	63.0	74.0	94.82	−0.29	2.62	2.29

^a^ λ_cut_: cutoff wavelength; ^b^ T_400_, T_450_: transmittance at 400 nm and 450 nm, respectively; ^c^ L*, a*, b*: color parameters, see Section 2.3.

**Table 4 materials-15-06346-t004:** Thermal properties of the FPI films.

PI	T_5%_ ^a^ (°C)	T_max1_ ^a^ (°C)	T_max2_ ^a^ (°C)	R_w750_ ^a^ (%)	T_g_ ^a^ (°C)	E′_50_ ^b^ (GPa)	E′_300_ ^b^ (GPa)	CTE ^c^ (×10^−6^/K)
FPI-1	513.7	525.3	579.4	56.6	367	1.21	0.39	30.6
FPI-2	515.8	529.9	580.5	55.0	360	3.56	1.19	29.9
FPI-3	517.7	532.0	584.1	55.5	365	3.91	1.19	27.1
FPI-4	520.3	533.1	584.1	57.2	357	4.25	1.29	25.1
FPI-5	522.3	532.3	584.4	56.8	361	4.53	1.38	24.2
FPI-6	524.4	535.4	595.4	58.5	367	9.87	3.20	23.4

^a^ T_5%_: temperature at 5% weight loss; T_max_: the rapid decomposition temperature; R_w750_: residual weight ratio at 750 °C in nitrogen; T_g,_: glass transition temperature detected by DMA measurements according to the peak temperatures of the loss modulus; ^b^ E′_50_, E′_300_: storage modulus of the FPI films at 50 °C and 300 °C, respectively; ^c^ CTE: linear coefficient of thermal expansion in the range of 50–250 °C.

## Data Availability

Data are contained within the article.

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
