# Peer review of "Synthesis and Properties of Optically Transparent Fluoro-Containing Polyimide Films with Reduced Linear Coefficients of Thermal Expansion from Organo-Soluble Resins Derived from Aromatic Diamine with Benzanilide Units"

_materials, 2022, doi:10.3390/ma15186346_

Round 1

Reviewer 1 Report

The manuscript entitled "Preparation and characterization of light-colored and transparent fluoro-containing polyimide films with reduced linear coefficients of thermal expansion from organo-soluble resins derived from aromatic diamine with benzanilide units" deals with the formulation of different kinds of light-colored fluoro-containing PI films and the characterization of their processability and thermal and optical properties. 

In my view, the topic of the work could be of interest for the readers of Materials and the manuscript is quite well written and organized. Besides, the experimental plan is well designed and the obtained results are clearly described in the paper.

For these reasons, i suggest the publication of the manuscript as it stands.

Author Response

Reply to the comments:

Thanks a lot for your positive evaluation on our work.

Reviewer 2 Report

The study aimed to prepare a series of wholly aromatic polyimide resins that containing fluorodianhydride and use them to fabricate thin films and study the optical and thermal properties of those films. The prepared resins were characterized by number of techniques and estimated their molecular weights and solubility in organic solvents. Films were prepared from these resins by casting method and their optical properties were studied. The results of the study were presented and discussed in a systematic manner as well as the method of preparation. Therefore, I recommend publishing the manuscript in the Journal of Materials.

My comments:

1.     The sample is less crystalline than the other samples according to the XRD spectra. Is there an explanation for that?

2.     The method of measuring the molecular weight of the resins should be written in detail in the characterization section.

Author Response

  1. Question: The sample is less crystalline than the other samples according to the XRD spectra. Is there an explanation for that?

Answer: The explanation was added in our revised manuscript as follows.

The amorphous nature for the current samples might be the synergistic effects of the bulky hexafluoroisopropylidene in the dianhydride moiety and the trifluoromethyl and methyl substituents in the diamine moiety.”.

  1. Question: The method of measuring the molecular weight of the resins should be written in detail in the characterization section.

Answer: The details on the measurement of the molecular weights for the PI resins were added in our revised manuscript as follows.

Number average molecular weight (Mn) and weight average molecular weight (Mw) of the PI resins dissolved in NMP containing 0.025 mol/L phosphoric acid were measured using a gel permeation chromatography (GPC) system (Shimadzu, Kyoto, Japan). HPLC grade NMP was used as the mobile phase at a flow rate of 1.0 mL/min. Polystyrene (Shodex, Type: SM-105, Showa Denko Co. Ltd., Tokyo, Japan) was used as the standard.”.